# Knocking Out *OsAAP11* to Improve Rice Grain Quality Using CRISPR/Cas9 System

**DOI:** 10.3390/ijms241814360

**Published:** 2023-09-21

**Authors:** Yihao Yang, Yi Zhang, Zixing Sun, Ziyan Shen, Youguang Li, Yifan Guo, Yuntong Feng, Shengyuan Sun, Min Guo, Zhi Hu, Changjie Yan

**Affiliations:** Jiangsu Key Laboratory of Crop Genetics and Physiology/Key Laboratory of Plant Functional Genomics of the Ministry of Education/Jiangsu Key Laboratory of Crop Genomics and Molecular Breeding, Yangzhou University, Yangzhou 225009, China; yihao.yang@yzu.edu.cn (Y.Y.); 007041@yzu.edu.cn (S.S.);

**Keywords:** CRISPR/Cas9, *OsAAP11*, grain quality, rice

## Abstract

The demand for rice grain quality, particularly in terms of eating and cooking quality, is increasingly concerning at present. However, the limited availability of rice-quality-related gene resources and time-consuming and inefficient traditional breeding methods have severely hindered the pace of rice grain quality improvement. Exploring novel methods for improving rice grain quality and creating new germplasms is an urgent problem that needs to be addressed. In this study, an amino-acid-transporter-encoding gene *OsAAP11* (*Os11g0195600*) mainly expressed in endosperm was selected as the target for gene editing using the CRISPR/Cas9 system in three *japonica* genetic backgrounds (Wuyungeng30, Nangeng9108, and Yanggeng158, hereafter referred to as WYG30, NG9108, and YG158). We successfully obtained homozygous *osaap11* mutants without transgenic insertion. Subsequently, we conducted comprehensive investigations on the agronomic traits, rice grain quality traits, and transcriptomic analysis of these mutants. The results demonstrate that loss of *OsAAP11* function led to a reduced amino acid content and total protein content in grains without affecting the agronomic traits of the plants; meanwhile, it significantly increased the peak viscosity, holding viscosity, and final viscosity values during the cooking process, thereby enhancing the eating and cooking quality. This study not only provides valuable genetic resources and fundamental materials for improving rice grain quality but also provides novel technical support for the rapid enhancement of rice grain quality.

## 1. Introduction

The improvement in people’s living standards has led to an increasing consumer focus on rice grain quality, making it a crucial breeding objective. The genetic mechanism underlying rice grain quality is highly intricate, resulting in sluggish progress in its improvement [1,2]. Therefore, it is extremely important and urgent to identify the key genes governing rice grain quality and devise a novel approach to enhance rice grain quality quickly.

Nitrogen is an essential element for the growth and development of rice, which determines the yield and quality of rice [3,4]. In soil, nitrogen exists in two forms: organic nitrogen (mainly ammonium and nitrates) and inorganic nitrogen (mainly amino acids, peptides, and proteins) [5,6,7]. Amino acids are an important organic nitrogen source and the main form of nitrogen transport in rice. The main sources of amino acids in plants include the direct uptake of proton co-transporters driven by H^+^-ATPase in roots from the soil, the conversion of nitrate or ammonium salts in roots, and synthesis occurring in leaves following the transport of xylem nitrate [8,9]. Subsequently, amino acids are loaded into the phloem and transported to the reservoir tissue through sieve tubes. During reproductive growth, organic nitrogen is exported through the seed coat to the interior of the seed, where it is stored as a storage protein in the endosperm or embryo/cotyledon [8,9]. Therefore, it is highly significant to investigate genes associated with amino acid biochemical pathways for enhancing rice yield and grain quality.

The transport of amino acids from the rhizosphere to the seed is facilitated by a series of membrane-localized amino acid transporters (AATs) [10]. In general, plant AATs can be divided into two major families: the amino acid/auxin permease (AAAP) superfamily and the amino acid polyamine and choline transport (APC) superfamily. Within the AAAP superfamily, there are eight subfamilies, including amino acid permeases (AAPs), lysine–histidine-like transporters (LHTs), proline transporters (ProTs), γ-aminobutyric acid transporters (GATs), auxin transporters (AUXs), aromatic and neutral amino-acid-like transporters (ANTs), amino acid transporters-like (ATLs) and amino acid vacuolar transporters (AVTs). The APC superfamily consists of subfamilies such as cationic amino acid transporters (CATs), bidirectional acid transporters (BATs), and L-type amino acid transporters (LATs) [11,12]. Currently, the accumulated knowledge about rice AATs is limited to the understanding of the amino acid permeases (AAPs) and lysine-histidine-like transporters (LHT) subfamilies, which are involved in amino acid absorption and transport from source to sink and ultimately affect rice yield and rice quality [13].

*OsAAP1* is expressed in the root tip, lateral root, leaf sheath, stem, axillary bud, young panicle, and leaf vascular bundle. It exhibits a wide range of amino acid transport capabilities and positively regulates tillering, panicle number, yield, and nitrogen use efficiency in rice [14,15]. *OsAAP3* is expressed in roots, stems, leaves, leaf sheaths, and panicles with a broad spectrum of amino acid transport abilities and is particularly notable for its basic amino acid transport ability. The knockout mutant prevents the excessive accumulation of amino acids at tiller nodes while significantly increasing the number of tillers and effective panicles per plant without affecting the nutritional quality of rice [14,16]. *OsAAP4* is primarily expressed in roots, leaves, leaf sheaths, stems, and ears, predominantly facilitating neutral amino acid transfer, which positively regulates tillering number, yield per plant, and nitrogen utilization in rice [17]. *OsAAP5* is mainly expressed in the root system including the tiller base as well as leaf sheaths and young panicles; it primarily transports basic and neutral amino acids. Knocking out this gene can effectively enhance biomass production along with increasing the number of tillers and yield per plant in rice [18]. The expression of *OsAAP6* is predominantly observed in the roots, glume vascular bundle, capital, internode, node, and endosperm. Overexpression of *OsAAP6* enhances the roots’ capacity to uptake amino acids, facilitating their transport through vascular bundle tissues to the endosperm. This ultimately improves the protein biosynthesis ability in the endosperm and indirectly influences key gene expressions related to starch biosynthesis [19,20,21]. *OsAAP7* and *OsAAP16* have a broad-spectrum ability to transport amino acids [14]. The expression of *OsAAP10* was relatively high in the stem and endosperm, whereas a significant reduction in grain protein content was observed upon *OsAAP10* knockout. Additionally, alterations were also detected in both the quantity and composition of starch [20]. Additionally, *OsLHT1* is expressed in various plant tissues including roots, stems, flag leaves, flag leaf sheaths, and young panicles. *OsLHT1* functions as a highly specific amino acid transporter with high affinity and demonstrates effective transport of aspartate, asparagine, and glutamate. Mutations in *OsLHT1* result in a significant reduction (40–66%) in root amino acid uptake along with impaired transportation of synthesized amino acids to the above-ground parts of rice. These alterations significantly impact growth and development processes while also leading to decreased biomass production and grain yield [21,22,23,24].

Although extensive research has been conducted on rice AATs genes, particularly the AAPs family genes, the majority of studies have focused on their impact on yield-related traits such as tillering number, grain number per panicle, and overall plant yield. Unfortunately, there has been limited attention given to their effect on quality-related traits. It is worth noting that numerous studies have indicated a potential negative correlation between rice yield and quality [1,2]. Therefore, it is crucial for us to address the urgent issue of enhancing rice quality without compromising agronomic traits. In this study, we selected *OsAAP11* as our target gene due to its relatively high expression in rice endosperm. By utilizing CRISPR/Cas9 technology, we performed gene knockout experiments in three different *japonica* rice cultivars to investigate their effects specifically on quality-related traits. This study aims to establish a foundation for comprehending the biological functions of amino acid transporter genes while offering theoretical support for improving rice quality.

## 2. Results

### 2.1. RT-qPCR Analysis of OsAAP11

In order to explore the expression characteristics of *OsAAP11* in various tissues of rice, particularly in grains at different stages of maturity, we employed the RT-qPCR technique to assess the expression levels of *OsAAP11* in roots, stems, leaves, leaf sheaths, and grains at distinct developmental stages across three diverse *japonica* rice cultivars (WYG30, NG9108, and YG158). Overall, *OsAAP11* exhibited similar expression patterns across different genetic backgrounds. It was relatively lowly expressed in rice roots, stems, leaves, and sheaths but highly expressed in grains at 5 days after flowering (DAF). Furthermore, gene expression decreased as the grains developed (Figure 1). These findings suggest that *OsAAP11* plays a pivotal role during early endosperm development.

### 2.2. Generation of osaap11 Mutants

In order to investigate the effect of *OsAAP11* (*Os11g0195600*) on rice grain quality, we employed the CRISPR/Cas9 system for targeted knockout, with the specific target site designed within the first exon of *OsAAP11* (44 bp downstream from ATG) (Figure 2A). Genome-wide off-target analysis was further conducted using CRISPR-GE (http://skl.scau.edu.cn/ (accessed on 22 January 2022)) and revealed a certain risk of off-target effects in the 5 ‘UTR regions of *Os07g0185300* and *Os01g0276700* (Figure 2B).

In the T_0_ generation, a total of 18, 20, and 20 independent transgenic plants were successfully obtained from three distinct genetic cultivars (WYG30, NG9108, and YG158). Among them, 15, 18, and 16 transgenic plants exhibited hygromycin resistance with positive rates of approximately 83%, 90%, and 80%. Subsequently, amplification sequencing was performed on regions with high off-target scores, but no gene editing events were detected (Appendix A). The T_0_ *hyg*-positive individual plants obtained were subsequently transferred to T_1_ lines and subjected to screening using *cas9* primers and *hygromycin* primers in order to identify transgene-free plants. Target site sequence amplification and sequencing were subsequently performed on the screened transgene-free plants. Finally, two types of homozygous mutations (−53 bp and −59 bp) were identified in the WYG30 background, while homozygous mutations (−4 bp and −1 bp) were found in both the NG9108 and YG158 backgrounds (Figure 2C–E). The aforementioned nucleotide mutations have the potential to cause frameshift mutations in amino acids.

### 2.3. Agronomic Traits in osaap11 Mutants

In order to assess the impact of gene functional loss on agronomic traits in plants, we first confirmed that potential off-targets were not edited in T_1_ plants and then conducted a systematic examination of the agronomic characteristics of mutants across different genetic backgrounds. The findings revealed a significant increase in grain length for mutant 11-WYG30-2. Conversely, mutant 11-NG9108-1 exhibited notable reductions in both plant height and grain length. Additionally, mutant 11-NG9108-2 displayed a substantial decrease in hundred-grain weight. Furthermore, mutant 11-YG158-1 demonstrated a significant decrease in plant height, while mutant 11-YG158-2 exhibited marked increases in both grain length and width (Figure 3, Table 1). Despite observing certain alterations within individual agronomic traits among these mutants, no discernible pattern was evident, suggesting that these phenomena are likely attributed to tissue culture effects.

### 2.4. Grain Protein Content in osaap11 Mutants

To investigate the impact of *OsAAP11* knockout on rice grain protein content, the grain powders of *osaap11* mutants with different genetic backgrounds were analyzed for total protein and storage protein contents using the Kjeldahl method. As depicted in Figure 4, compared with the wild type, *osaap11* mutants in the WYG30 background exhibited significant reductions of 17.9% and 19.3% in total protein content, while those in the NG9108 background showed decreases of 3.3% and 4.3%. In the YG158 background, *osaap11* mutants displayed declines of 13.9% and 14.6%. The results demonstrate that *OsAAP11* knockout can significantly reduce the total grain protein content to a similar extent within the same genetic background, albeit with slight variations observed across different genetic backgrounds. Notably, WYG30 exhibited the most pronounced reduction, approaching 20%, while NG9108 displayed the smallest decrease, both falling below 5%.

Further analysis of the contents of four kinds of storage protein revealed that most mutants exhibited lower levels of albumin, globulin, and prolamin compared with the wild type, without achieving statistical significance. Conversely, the glutelin content was substantially reduced in all mutants relative to the wild type to a similar degree as that observed for total protein (Figure 4). The above results indicate that the loss of *OsAAP11* function led to a partial reduction in the rice storage protein, and the glutelin content was the primary factor.

### 2.5. Amino Acid Content in osaap11 Mutants

It has been shown that members of the AAPs family are responsible for the transmembrane transport of specific amino acids in plants [13]. To investigate the effect of *OsAAP11* knockout on the amino acid contents in rice grains, the amino acids were extracted and further determined. As depicted in Figure 5A,C,E, the content of each amino acid decreased to varying degrees in the mutants in the three genetic backgrounds, particularly the contents of aspartate, glutamate, valine, and arginine with high contents. The total amino acid content significantly declined across different backgrounds, which was similar to the trend observed for rice protein content. The decrease rate was smallest (4.4% and 7.6%) for mutants in the NG9108 background, while it exceeded 10% for the WYG30 and YG158 backgrounds, respectively (Figure 5B,D,F). Furthermore, *OsAAP11* knockout led to a reduction ranging from 0.5 to 1.8 mg/g for basic amino acids; from 0.8 to 4.2 mg/g for acidic amino acids; and from 0.4 to 7.8 mg/g for neutral amino acids (Figure 5B,D,F). In conclusion, the loss of *OsAAP11* function mostly affects neutral amino acids and, in general, the contents of amino acids in rice grains.

### 2.6. Starch Content and Physicochemical Properties of osaap11 Mutants

In order to investigate the potential impact of *OsAAP11* on the physicochemical properties of rice starch, we initially assessed the total starch and amylose contents in each mutant. As depicted in Figure 6A,B, the total starch contents in 11-WYG30-2, 11-YG158-1, and 11-YG158-2 exhibited significant increases compared with their corresponding wild types, with increments of 1.7%, 3.3%, and 3.7%, respectively. Furthermore, mutants 11-WYG30-1 and 11-WYG30-2 displayed notable decreases in amylose contents of 2.1% and 1.8%, respectively. The most probable cause of this phenomenon is the rice endosperm’s role in maintaining the carbon and nitrogen balance. These findings suggest that the absence of *OsAAP11* not only affected the amino acid and protein contents in rice grains but also played a crucial indirect regulatory role in determining starch quantity and composition.

Starch viscosity analysis serves as an effective tool for evaluating rice eating and cooking quality by simulating the rice cooking process to measure starch gelatinization characteristics [25]. The following figure illustrates that, in comparison with the corresponding wild types, nearly all mutants exhibited higher peak viscosity, holding viscosity, and final viscosity while demonstrating reduced setback values, indicating that loss of *OsAAP11* function is highly likely to influence the palatability of rice (Figure 6C).

### 2.7. Taste Scores of osaap11 Mutants

In order to investigate the impact of *OsAAP11* gene mutation on the eating and cooking quality of cooked rice, the taste scores of all mutant types were subsequently assessed [26]. The findings reveal a significant enhancement in eating and cooking quality for all mutants across the three genetic backgrounds, with taste score increases ranging from 3.9% to 17.2% and an average increment of 9.8%. Notably, YG158 exhibited the most pronounced effect, resulting in a more than 10% improvement in the taste score for cooked rice, while WYG30 demonstrated a moderate improvement effect, and NG9108 showed the least enhancement (Figure 7).

### 2.8. Transcriptomic Analysis of Genes Related to Protein and Starch Biosynthesis in osaap11 Mutants

In order to further investigate the regulatory mechanism of *OsAAP11* on rice quality traits, RNA-Seq analysis was conducted on 11-WYG30-1 and its background parent WYG30. A total of 1596 differentially expressed genes were identified based on two criteria: a fold change greater than |log2(Fold change)| > 1 and a significance level of *p*-value < 0.05. Among these genes, 1108 were significantly upregulated, while 547 were significantly downregulated. Gene Ontology (GO) functional enrichment analysis revealed that these differentially expressed genes were primarily associated with nutrient reservoir activity, cytoplasmic ribosomes, aleurone grain, transmembrane transport activity, intrinsic components of the plasma membrane, and substrate-specific transmembrane transporter activity (Figure 8A). Kyoto Encyclopedia of Genes and Genomes (KEGG) pathway enrichment analysis indicated that the differentially expressed genes mainly participated in ribosome, photosynthesis-antenna protein, flavonoids, and flavonols biosynthesis; alanine, aspartate, and glutamate metabolism; taurine and hypotaurine metabolism; nitrogen metabolism; starch and sucrose metabolism; betaine biosynthesis; glycosaminoglycan degradation; and so on (Figure 8B).

To investigate whether alterations in storage protein and starch accumulation were reflected by changes in mRNA levels, we further examined the expressions of 49 key genes associated with storage substances in endosperm at 5 DAF. This encompassed 36 genes related to grain protein synthesis and 13 genes related to grain starch metabolism. The findings revealed that most genes involved in storage protein synthesis were downregulated, particularly *OsGluA1-OsGluB5*, *RM1*, and *Glb1*, which exhibited high endogenous expression levels. The genes associated with starch metabolism remained largely unaltered, while a few genes involved in starch synthesis exhibited slight upregulation (Figure 8C).

## 3. Discussion

Numerous studies have extensively investigated the impact of grain protein content on the eating and cooking quality of rice. Xie et al. conducted a study where they introduced DTT into the cooking water to enzymatically hydrolyze proteins [27]. This resulted in reduced rice hardness and a significant increase in viscosity in non-glutinous rice varieties. These findings suggest that rice protein not only affects texture but also plays a crucial role in determining rice viscosity. In another study, Yang et al. analyzed the levels of total protein and storage proteins in nearly 100 *japonica* rice cultivars. They then conducted correlation analyses with the taste scores of the cooked rice. The results revealed a strong negative correlation between total protein content, glutelin content, and taste score, with correlation coefficients of −0.89 and −0.57, respectively [28]. Furthermore, Park et al. utilized a collection of RIL populations to identify a stable rice grain protein content QTL*qPro9* on chromosome 9. Interestingly, this QTL was found to be closely associated with a QTL*qTV9* related to taste score, with an additive effect that was in the opposite direction [29]. The aforementioned studies directly or indirectly indicate that high levels of grain protein negatively impact the taste characteristics of rice overall.

Under normal circumstances, the grain protein content in high-quality rice is often lower than 7% [28]. However, in actual production practice, the grain protein content in ordinary rice cultivars is generally high due to the high nitrogen application level and variety characteristics. This high grain protein content seriously affects the eating and cooking quality of rice. Previous genetic studies have primarily concentrated on identifying QTLs associated with rice protein content. Among these QTLs, only *qPC1* and *qGPC-10* have been successfully cloned [19,30]. Therefore, the scarcity of gene resources and methods available for improving rice grain protein-related traits severely hampers the further enhancement of rice grain quality.

Plant AAPs regulate the process of organic nitrogen release from the parental grain coat to the grain endoplasm, thereby influencing the storage protein content in either the endosperm or embryo/cotyledon. In Arabidopsis thaliana, *AtAAP1* is involved in transporting amino acids to the embryo, and the amino acid content in the endosperm of the mutant is significantly decreased, and the protein level is also significantly decreased [31]. *AtAAP8* can transport amino acids for early embryonic growth, and its mutant production is significantly reduced [32]. The T-DNA insertion line of *AtAAP2* also results in a significant reduction in protein content within dried seeds [33]. In peas, *VfAAP1* overexpression lines could significantly increase seed protein content under different nitrogen levels [34]. The *OsAAP6* gene in rice plays a positive role in regulating rice protein content. Overexpression of *OsAAP6* enhances the root’s ability to absorb certain amino acids, promotes the expression of key genes related to storage proteins, and ultimately results in an expanded proteome and increased rice protein content [19]. Therefore, regulating the expression of rice *AAPs* presents a viable approach to improve rice protein content and improve rice grain quality.

In this study, we observed a high expression of *OsAAP11* in the endosperm of rice 5 days after flowering, indicating its crucial role in early rice endosperm development. Subsequently, CRISPR/Cas9 gene-editing technology was employed to knock out *OsAAP11* in three different genetic backgrounds. The results revealed significant reductions in amino acid and total protein contents across almost all mutants, accompanied by increased peak viscosity, holding viscosity, final viscosity, and improved rice taste scores. Remarkably, these improvements were achieved without substantial changes in agronomic traits. This study not only provides valuable genetic resources for enhancing rice quality but also presents a viable approach for rapid improvement.

## 4. Materials and Methods

### 4.1. Experimental Materials and Plant Method

The experimental materials used in this study were the Wuyungeng 30, Nangeng 9108, and Yanggeng 158 rice varieties suitable for cultivation in Jiangsu Province. These materials were cultivated in Yangzhou, Jiangsu Province, China, and Lingshui, Hainan Province, China. Each genotype was represented by a single plant with two rows of 10 plants per row. The spacing between plants was maintained at 20 cm × 20 cm. Standardized water and fertilizer management practices were followed.

The PC1300-Cas9 knockout vector and SK-gRNA intermediate vector were generously provided by Dr. Kejian Wang, a renowned researcher at the China Rice Research Institute.

### 4.2. Target Site Design and Mutation Detection

The coding sequence of *OsAAP11* (*Os11g0195600*) was obtained from the RGAP website (http://rice.plantbiology.msu.edu/ (accessed on 22 January 2022)). In the first exon, a targeted gene sequence was designed. The target sequence for *OsAAP11* was GGAGGCGGGGATGATGGTGGG, which is located 44 bp downstream of the transcription start codon ATG. The CRISPR—GE (http://skl.scau.edu.cn/offtarget/ (accessed on 22 January 2022)) website was utilized for the analysis of off-target effects for the target sequence. The vector construction method and process were carried out with reference to the literature [20]. The NCBI (http://www.ncbi.nlm.nih.gov (accessed on 27 January 2022)) database was utilized to identify the appropriate design primer *OsAAP11*-jc-F and *OsAAP11*-jc-R sequences (*OsAAP11*-jc-F: CAACCCTACACGCCTCTCTC; *OsAAP11*-jc-R: AGGTTGGCGTACTGGATGAC). Subsequently, the CTAB (cetyltrimethylammonium bromide) method was employed for DNA extraction from transgenic plants, followed by PCR amplification sequencing to determine the types of mutations.

### 4.3. Transgenic Component Detection

The transgenic T_0_ plants in our study were obtained using the agrobacterium-mediated method at Biorun Bio-Company, Wuhan, China (https://plant.biorun.com/ (accessed on 10 May 2022)). In the T_1_ generation population, DNA was sampled and extracted from a single plant, and exogenous transgenic components were detected using specific primers targeting the *cas9* coding gene (Cas9-F and Cas9-R) as well as the *hygromycin* gene (Hyg-F and Hyg-R) (Appendix A). The absence of transgenic components in plants was confirmed by the absence of corresponding bands.

### 4.4. Analysis of Protein Contents and Amino Acid Contents in Rice Grains

The total nitrogen content in rice flour and the protein content of each component were analyzed with the kjeltec automatic nitrogen analyzer (FOSS, Hilleroad, Denmark) and the protein content was calculated using 6.25 as the nitrogen conversion coefficient.

The steps for extracting 4 kinds of storage protein were as follows: An amount of 1 g of rice flour was weighed. Albumin was extracted using 1 mL of ddH_2_O. Globulin was extracted using 1 mL of a solution containing 50 mM KH_2_PO_4_, 0.5 mM NaCl, and 1 mM EDTA-2Na (pH = 6.8). Alcohol-soluble protein was extracted using 1 ml of 75% ethanol. Glutenin was extracted from the EDTA-2Na solution using a mixture of 0.1 mM NaOH and 1 mM concentration. Each component protein was shaken at a temperature of 4 ℃ for a duration of six hours, followed by centrifugation at a speed of 10,000 rpm for fifteen minutes, which was repeated three times.

An amount of 10 mg of rice flour was weighed into a 2 mL spiral cap tube and 300 nM of L-(+)-Norleucine was added for subsequent correction purposes. An amount of 500 µL of 6 N HCl was hydrolyzed in an oven at 110 °C for 24 h, followed by centrifugation at 13,000 rpm for 10 min and complete drying in a water bath at 57 °C. The residues were dissolved in an amino acid diluent and centrifuged at 10,000 rpm for another 10 min. The supernatant was filtered using a nylon membrane syringe with a pore size of 0.45 μm and transferred to an automatic sample vial for analysis using an automatic amino acid analyzer (Biochrom30+, Cambridge, England).

### 4.5. Analysis of Amylose and Total Starch Contents in Rice Grains

The amylose content determination was carried out as follows: Amounts of 0.1 g of the sample to be tested and a standard sample were weighed, and then 1 mL of ethanol solution was added into a 10 mL centrifuge tube. Next, 9 mL of a 1 mol/L NaOH solution was added into the same centrifuge tube and shaken vigorously. The tube was wrapped with a sealing film and heated in boiling water for 10 min. It was allowed to cool naturally, and then 1 mL of the mixture was diluted into another 10 mL centrifuge tube. An amount of 1 mL of acetic acid solution and 2 mL of iodine reagent were added to this new tube. It was filled up with water until it reached a total volume of 10 mL, shaken well, and let sit for another 10 min. Finally, the spectrophotometer (BioTek, Winooski, Vermont, United States) was zeroed using sodium hydroxide solution before measuring the absorbance at a wavelength of 720 nm.

The determination of total starch content involved the preparation of an anthrone–sulfuric acid solution by dissolving 0.4 g of anthrone in 100 mL of H_2_SO_4_. Amounts of 0.1 g of the sample to be tested and a standard sample were weighed and placed in separate 50 mL centrifuge tubes. Then, 30 mL of 52% perchloric acid was added to each tube for complete shaking and dissolution. The residue at the pipe mouth was washed with water, followed by adjusting the volume to 50 mL and diluting it by a factor of 25. Additionally, the standard starch solution was further diluted into concentrations of 12.5%, 25%, 50%, 75%, and 100%. Each sample (1 mL) was transferred into a separate 10 mL centrifuge tube, followed by adding 6 mL of the anthrone–sulfuric acid reagent for thorough mixing. After cooling, the tubes were placed in boiling water for 5 min before being taken out to cool down to room temperature. Finally, the absorbance at 640 nm was measured using a spectrophotometer after zero adjustment with distilled water.

### 4.6. Analysis of Starch Viscosity Characteristics

In accordance with the AACC (American Association of Cereal Chemists) procedures, the RVA TecMaster (Perkinelmer, Sydney, Australia) and its accompanying TCW (3.17.3.509) (Thermocline for Windows) software was utilized for the analysis. An amount of 3 grams of rice flour with a water content of 13% was weighed, followed by the addition of 25 g of water. The mixture was thoroughly stirred in an aluminum can before being subjected to analysis using the machine. The procedure consisted of the following cycles: 1 min at 50 °C, 2.5 min at 95 °C, 1.4 min at 50 °C, and a total duration of 3.8 min, including both heating and cooling processes. Throughout this period, the RVA (Rapid Visco Analyser) blades rotated at an initial speed of 960 rpm for the first ten seconds and subsequently maintained a constant speed of 160 rpm for the remaining time interval. The RVA spectral characteristic values were expressed through parameters such as peak viscosity, holding viscosity, final viscosity, breakdown value (peak viscosity minus holding viscosity), setback value (final viscosity minus peak viscosity), as well as other relevant characteristics.

### 4.7. Analysis of Rice Taste Scores

An amount of 30 g of milled rice was cleaned with tap water, water was added according to a 1:1.3 ratio, and it was sealed with filter paper and rubber band, soaked for 30 min, and steamed in a pot for 30 min, the power was turned off, and it was stewed for 10 min. When finished, the rice was turned with a spoon to prevent it from sticking to the sides of the pot. Then, the air in the cooling box was blown for 10 min, the lid of the matching can was closed, and then it was left to stand for 90 min, the machine was preheated in advance for 30 min, the software was opened, and the Japanese *Japonica* rice mode was entered for calibration. The cooled rice was weighed to 8.00 g in a metal ring, with positive and negative pressure for 10 s, and the STA1A eating instrument (Sasaki Corporation, Aomori, Japan) was used to determine the taste score of the rice [26]. The working principle of this instrument is converting various physicochemical parameters of rice into “taste” scores based on correlations between the near-infrared reflectance (NIR) measurements of key constituents (e.g., amylose, protein, moisture, and fat acidity) and preference sensory scores.

### 4.8. Investigation of Main Agronomic Characteristics of Rice Plants

The main panicles of 5 mutants and 5 wild-type rice plants were selected at the mature stage for the investigation of height, tiller, number of grains per panicle, grain length, grain width, 100-grain weight, and fertility. Grain length and width were analyzed using the seed-measuring instrument (Model SC-G, Wanshen, Hangzhou, China).

### 4.9. Transcriptome Analysis

Total RNA was isolated from the endosperm at 15 DAF of WYG30 and 11-WYG30-1, using TRIzol reagent (Invitrogen, Shanghai, China) and purified using an RNeasy Plant Mini Kit (Qiagen, Venlo, Netherlands). Each sample contained over 25 grains, with 3 biological replicates, and was used for paired-end library construction. Sequencing was performed on an Illumina platform of the Gene Denovo company, China (https://www.genedenovo.com/ (accessed on 28 May 2023)). HTSeq v0.9.1 was used to count the read numbers mapped to each gene. The DESeq R package (1.18.0) was used for analyzing the differentially expressed genes. The Go functional enrichment and KEGG pathway enrichment analyses were performed using the OmicShare tools, a free online platform for data analysis (https://www.omicshare.com/tools (accessed on 28 May 2023)).

### 4.10. RNA Extraction, cDNA Preparation, and qRT-PCR

Total RNA was extracted from endosperm at 15 DAF using an RNA extraction kit (Tiangen, Beijing, China). First-strand cDNA was synthesized using a reverse transcription kit (Vazyme, Nanjing, China). Quantitative reverse transcriptase (qRT-PCR) was performed with a CFX96 Real-Time PCR System (Bio-Rad, Hercules, CA, USA) using an SYBR qPCR Master Mix (Vazyme, Nanjing, China). All assays were performed with at least three biological replicates. The rice actin gene served as the internal control to normalize gene expression. All primers are listed in Appendix A.

### 4.11. Data Analysis

The experiment was conducted with three biological replicates for each sample. Microsoft Excel 2016 was utilized for data collection, while the statistical analysis software SPSS 15.0 was employed for variance analysis. All experimental data are presented as means ± SD.

## Figures and Tables

**Figure 1 ijms-24-14360-f001:**
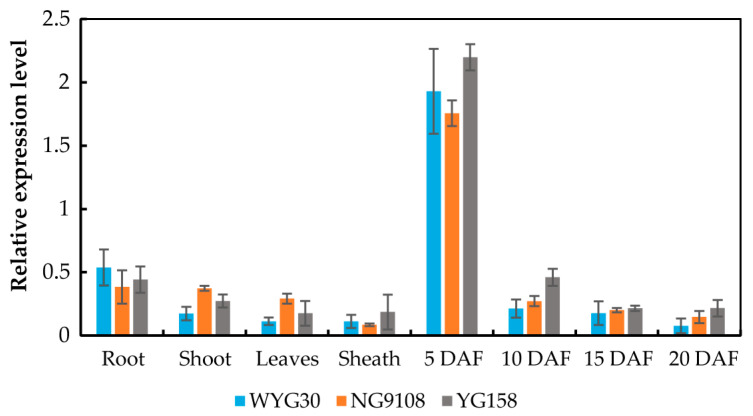
RT-qPCR analysis of *OsAAP11* expression.

**Figure 2 ijms-24-14360-f002:**
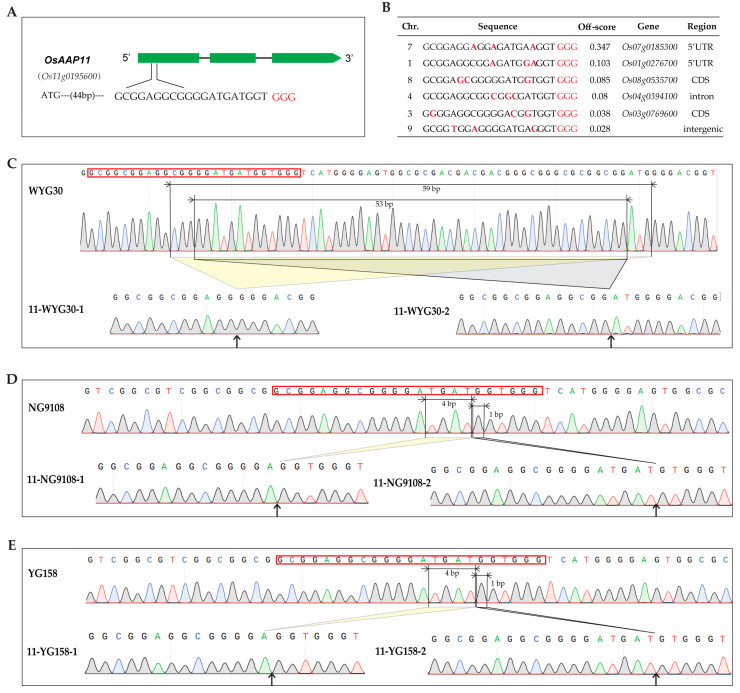
Generation of *osaap11* mutants using CRISPR/Cas9 system. (**A**) Schematic map of the genomic region of *OsAAP11*. The protospacer-adjacent motif (PAM) is indicated by red letters. (**B**) Off-target sequence analysis. The different nucleotides are marked in bold. (**C**) Mutation detection in T_1_. Sanger chromatogram of intact WYG30 and mutated *OsAAP11* target motif. 11-WYG30-1 carried 59 bp deletion and 11-WYG30-2 carried 53 bp deletion. (**D**) Mutation detection in T_1_. Sanger chromatogram of intact NG9108 and mutated *OsAAP11* target motif. 11-NG9108-1 carried 4 bp deletion and 11-NG9108-2 carried 1 bp deletion. (**E**) Mutation detection in T_1_. Sanger chromatogram of intact YG158 and mutated *OsAAP11* target motif. 11- YG158-1 carried 4 bp deletion and 11-YG158-2 carried 1 bp deletion. The sequence marked by a red frame indicates the target motif. Arrows indicate fragments deleted from the WT sequence and the corresponding ligation points in the mutant alleles.

**Figure 3 ijms-24-14360-f003:**
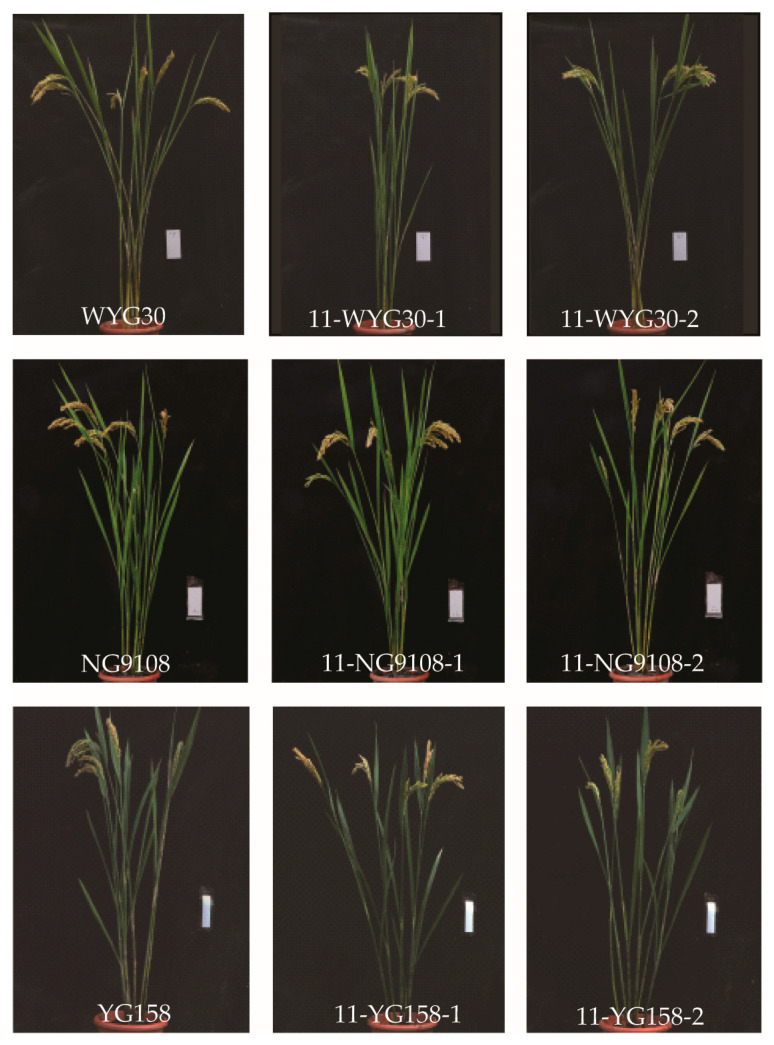
Whole plant of the wild-type and homozygous mutants. Scale = 10 cm.

**Figure 4 ijms-24-14360-f004:**
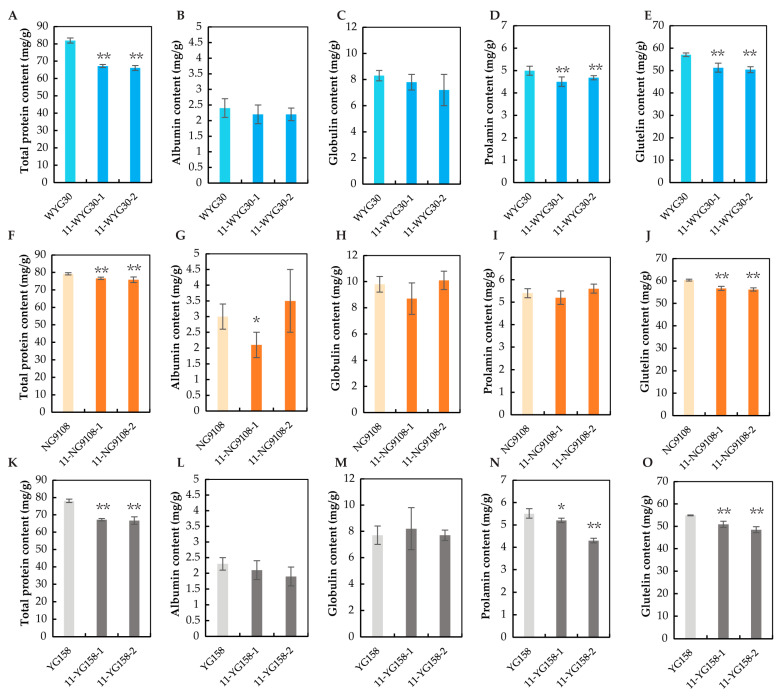
Grain protein contents in wild-type and *osaap11* mutants. (**A**,**F**,**K**) The total protein content in the wild type and the mutants. (**B**,**G**,**L**) The albumin content in the wild type and the mutants. (**C**,**H**,**M**) The globulin content in the wild type and the mutants. (**D**,**I**,**N**) The prolamin content in the wild type and the mutants. (**E**,**J**,**O**) The glutelin content in the wild type and the mutants. *p*-value < 0.05 * and < 0.01 **, calculated using independent-samples *t*-test.

**Figure 5 ijms-24-14360-f005:**
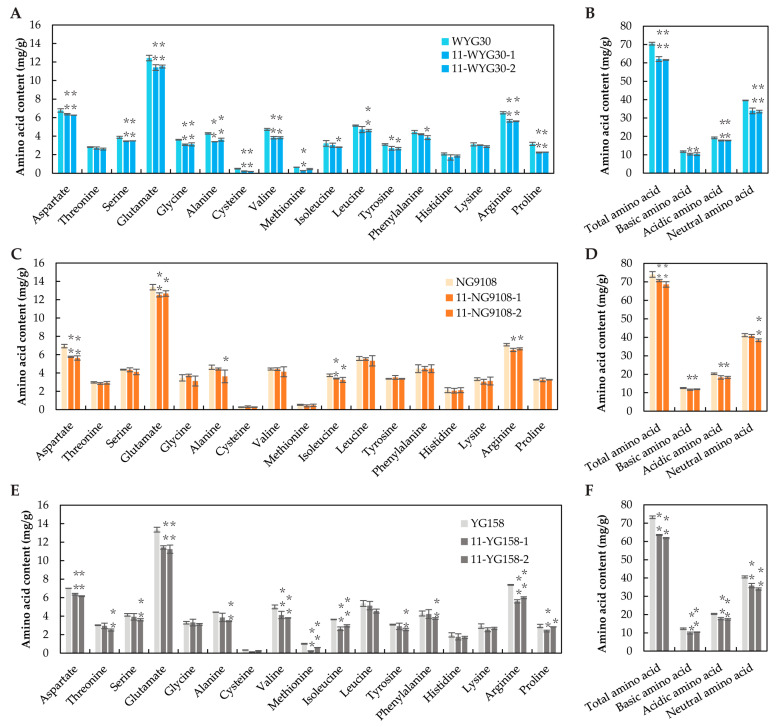
Comparison of amino acid contents between the wild-type and homozygous mutant lines. (**A**,**C**,**E**) The amino acid contents in the wild types and the mutants. (**B**,**D**,**F**) The total, basic, acidic, and neutral amino acid contents in the wild types and the mutants. *p*-value < 0.05 * and < 0.01 **, calculated using independent-samples *t*-test.

**Figure 6 ijms-24-14360-f006:**
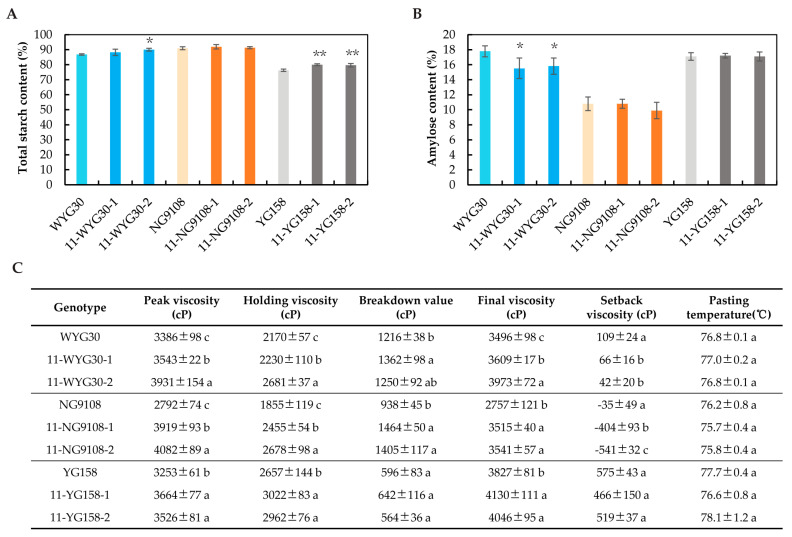
Starch contents and physicochemical properties of osaap11 mutants. (**A**) Comparison of total starch contents between the wild-type and homozygous mutant lines (100% = 1 g rice flour). (**B**) Comparison of the amylose contents between the wild-type and homozygous mutant lines (100% = 1 g rice flour). (**C**) RVA (Rapid Visco Analyser) profile of the wild-type and homozygous mutant lines. *p*-value < 0.05 * and < 0.01 **, calculated using independent-samples *t*-test. Different lowercase letters represent significant differences at *p* < 0.05 determined using one-way ANOVA with Duncan’s multiple range test.

**Figure 7 ijms-24-14360-f007:**
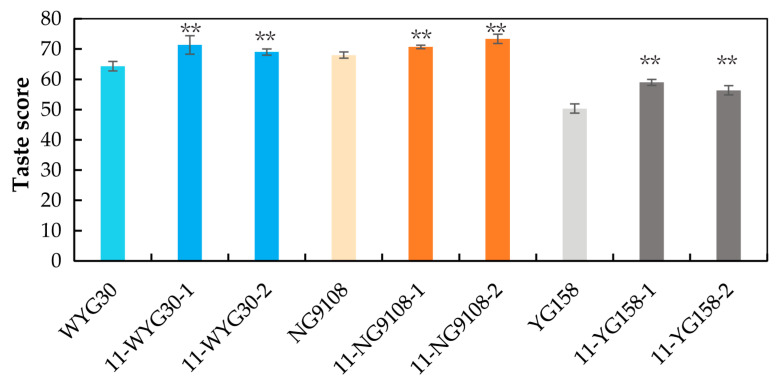
Taste scores of the wild-type and homozygous mutant lines. *p*-value < 0.01 **, calculated using independent-samples *t*-test.

**Figure 8 ijms-24-14360-f008:**
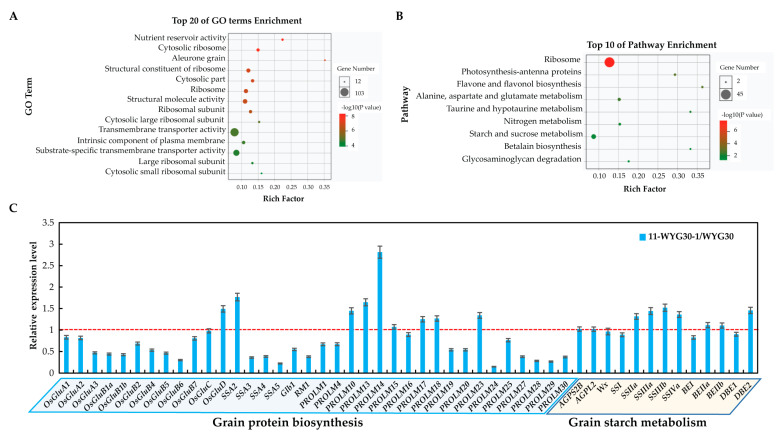
Transcriptomic analysis of genes related to protein and starch biosynthesis in *osaap11* mutants. (**A**,**B**) GO functional and KEGG pathway enrichment analyses of the DEGs. (**C**) The relative expression analysis for the genes related to protein and starch biosynthesis in *osaap11* mutants.

**Table 1 ijms-24-14360-t001:** Analysis of agronomic characteristics of the wild-type and homozygous mutant lines.

	Plant Height (cm)	Tiller Number	Grain Number/Panicle	Fertility (%)	Grain Length (cm)	Grain Width (cm)	Grain Weight/100 (g)
WYG30	100.9 ± 1.3a	10.0 ± 0.9a	192.2 ± 16.0a	93.1 ± 1.4a	7.21 ± 0.05bc	3.65 ± 0.04a	2.77 ± 0.14a
11-WYG30-1	100.9 ± 2.7a	10.0 ± 1.0a	194.3 ± 14.0a	89.2 ± 0.9a	7.12 ± 0.05c	3.70 ± 0.02a	2.66 ± 0.10a
11-WYG30-2	100.2 ± 1.9a	8.3 ± 1.0a	187.0 ± 20.5a	88.9 ± 2.4a	7.26 ± 0.10a	3.68 ± 0.05a	2.70 ± 0.12a
NG9108	94.1 ± 2.1a	10.5 ± 1.4a	176.8 ± 10.7a	90.5 ± 1.7a	7.11 ± 0.10a	3.85 ± 0.03a	2.78 ± 0.07a
11-NG9108-1	85.0 ± 3.3b	9.7 ± 0.6a	183.3 ± 5.9a	88.2 ± 2.5a	7.08 ± 0.03b	3.72 ± 0.07a	2.71 ± 0.10ab
11-NG9108-2	93.1 ± 1.8a	10.7 ± 2.1a	174.3 ± 11.1a	88.7 ± 1.9a	7.11 ± 0.10a	3.79 ± 0.06a	2.67 ± 0.05b
YG158	76.9 ± 3.5a	8.3 ± 0.6a	130.2 ± 6.9a	91.7 ± 2.0a	7.06 ± 0.29b	3.64 ± 0.04b	2.78 ± 0.12ab
11-YG158-1	70.9 ± 2.2b	8.7 ± 1.5a	121.3 ± 9.0a	89.3 ± 1.4a	7.09 ± 0.06b	3.64 ± 0.05b	2.71 ± 0.06b
11-YG158-2	76.6 ± 2.1a	8.7 ± 2.1a	134.3 ± 11.6a	89.7 ± 4.8a	7.31 ± 0.12a	3.72 ± 0.03a	2.82 ± 0.06a

Different lowercase letters represent significant differences at *p* < 0.05 determined using one-way ANOVA with Duncan’s multiple range test.

## Data Availability

Not applicable.

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
