# Peer review of "Knocking Out OsAAP11 to Improve Rice Grain Quality Using CRISPR/Cas9 System"

_ijms, 2023, doi:10.3390/ijms241814360_

Round 1
Reviewer 2 Report
The study from Yang and colleagues elucidates the consequences for a targeted knock out of amino acid transporter OsAAP11 and gives hints about gene expression as well as the content of sugar, proteins and amino acids. Nonetheless, some improvements are necessary before publication:
General comments:
1. The authors should check the writing of proteins and genes, especially when they describe mutants (e.g. lines 169 and 171, but also other regions of the paper)
2. All abbreviations have to be explained at their first use (e.g. RVA, AACC, TCW, CTAB)
3. The quality of all figures must be improved, because they are all blurred. Especially in Fig. 2 it was very difficult, to see the PAM (Fig. 2B) and the sequences in the sequencing chromatograms (Fig. 2C)
4. In the figure captions the authors should divide the subfigures more clearly, sometimes it was not easy to grab, what is described (see also detailed comments) - In general, the figure captions do not contain sufficient details. The standard requirements in scientific literature is that figure captions have to present all details required for full comprehension of the figure by itself, i.e. without information additionally provided in the context.
5. Grasses have no seeds but fruits, i.e. grains or caryopses. The text is to be corrected accordingly.
6. Write wild-type throughout the entire manuscript.
7. Leaf a blank between numeric values and unit.
Detailed comments:
Lines 125-130: which gene ID has AAP11? Other IDs are mentioned, but the one for AAP11 is not
Line 131: What is meant by positive single strains. Is it hpt-positive or mutated plants?
Line 132: Provide the number of putative T0 plants generated and analyzed.
Lines 136f: what is meant by negative plants? T-DNA free ones?
Lines 138f: “two types of mutations” is repetitive
Line 139: Fig. 2A describes the mutants already; 2C shows them in a larger context
Line 142/ Fig. 2C: find a better way to highlight the deletions together with sequencing peak pictures; Better show a wild-type chromatogram above to which modifications can be visually referred (e.g. Fig.3b in https://doi.org/10.1186/s12870-020-02454-9)
Line 145: The M in PAM stands for motif, so the word motif following PAM is repetive.
Line 151: There is no Table 3 existing in the paper.
Line 154: Thousand grain weight is explained (also later in line 411), wheras 100 grain weight is used in Table 1
Lines 158f: The phenotype was investigated in T1. Potential off-targets also need to be analyzed in T1, because they may have been induced in T0 only after tissue sampling for analyses. Undetected off-target mutations could be one reason for the different phenotypes found.
Line 160 / Fig. 3: quality of pictures and the text in the pictures is not acceptable.
Line 162 / Table 1: Be consistent with the number of digits after the comma. The meaning of the significance-associated letters have to be explained in the context of the table's title.
Lines 166ff: belongs to the method part
Lines 174ff: combine with the previous sentence, otherwise repetitive
Line 185 / Caption of Fig. 4: add information about significance symbols (p value <0.05 *, <0.01 **, <0.001 ***)
Lines 188f: add a reference.
Lines 190ff: “amino acid content” and “content” is repetitive
Line 191: which components were tested? a hint about different components is missing in the text and in Fig. 5. Please clarify and add component-specific results (e.g. in the supplement)
Line 200: suggestion: “In conclusion, the loss of OsAAP11 function mostly effects neutral amino acids and in general the content of amino acids in rice grains.”
Line 206: “content” is repetitive
Lines 205-213: Explain the extremely different behaviour of the different genotypes, although they perform similar at amino acid and protein levels
Lines 216f / See also lines 392ff: Explain, what is meant. Which parameter is shown in the Fig. 6, and where are the data for the other parameters explained in the Material and Methods section?
Line 218: delete “ultimate”
Line 220 / Fig. 6 / see also Line 377: To which criteria amylose and starch content are related? To what is the 85% starch content for WYG30 related? In other words, what is 100%?
Line 221 / Fig. 6 caption: Explain A,B,C,D,E. What is RVA? Add information given in lines 215f to the caption.
Lines 225-232: what is a taste value, i.e. what is exactly measured? , what is the difference to the 'eating value' referred to in line 405, or is it the same? provide a reference.
Line 333: Information about the rice transformation procedure is missing. Also provide the type and number of explants used to generate transgenic/mutated plants.
Line 338: Information about the analysis of primary transgenic plants (T0) is missing here.
Line 340: add primer sequences for Cas9 and hpt primers
Line 351: “1 mm concentration” is to be corrected
Lines 404f: what is the STA/A eating instrument doing? Briefly describe and give a reference.
Please improve the language and avoid word repetitions (see also detailed comments).
Round 2
Reviewer 2 Report
The authors have improved their manuscript, but following points still have to be changed before publication:
1. Fig 2D/E: better highlight 1bp deletion on the G left of the marked one, because Cas9 cutting positions are 3 to 4 bp upstream of PAM and not as you show 2 bp before PAM.
2. Fig 2 caption:
- Change (A) to: "Schematic map of the genomic region of OsAAP11, the protospacer-adjacent motif (PAM) is indicated by red letters."
- Describe more clear which genotypes are shown in C, D and E
3. Fig 4, 5, 6, 7,8 figures now larger, but quality has to be improved! In fact, old version of Fig. 8 was more sharp!
4. Fig 6 caption: abbreviation RVA should be explained here, otherwise readers might not understand the meaning!
5. Comment and Response 32: please add this important information to the manuscript (100 % = 1g rice flour)
6. Chapter 2.7 and related parts: based on new added reference 26 I suggest to rename "taste value" by "taste score" which corresponds better to the reference given.
